Feature selection using a multi-strategy improved parrot optimization algorithm in software defect prediction

Fei Qi 1 2 feiqixia@163.com
Yin Guisheng 1
Sun Zhian 2
1 College of Computer Science and Technology, Harbin Engineering University , Harbin , China
2 Jiangsu Automation Research Institute , Lianyungang , China
Kong Xiangjie
Electronic publication date: 2025 Apr 16
Publication date: 2025
Volume: 11
Electronic Location ID: e2815
Received 2024 Oct 22; Accepted 2025 Mar 19
Copyright: © 2025 Fei et al.
Copyright year: 2025
Copyright holder: Fei et al.
License: This is an open access article distributed under the terms of the Creative Commons Attribution License, which permits unrestricted use, distribution, reproduction and adaptation in any medium and for any purpose provided that it is properly attributed. For attribution, the original author(s), title, publication source (PeerJ Computer Science) and either DOI or URL of the article must be cited.
License URL: https://creativecommons.org/licenses/by/4.0/

Keywords: Software defect prediction, Parrot optimization, Machine learning, Stacked ensemble, Imbalanced datasets

Funding: The authors received no funding for this work.

==============================
Software defect detection is a critical research topic in the field of software engineering, aiming to identify potential defects during the development process to improve software quality and reduce maintenance costs. This study proposes a novel feature selection and defect prediction classification algorithm based on a multi-strategy enhanced Parrot Optimization (PO) algorithm. Firstly, to address the limitations of the original Parrot Optimization algorithm, such as strong dependence on the initial population, premature convergence, and insufficient global search capability, this article develops a multi-strategy enhanced Parrot Optimization algorithm (MEPO). Experiments conducted on eight benchmark test functions validate the superior performance of MEPO in terms of convergence speed and solution accuracy. Secondly, to mitigate the adverse impact of irrelevant features on model performance in traditional software defect prediction methods, this study introduces a binary multi-strategy enhanced Parrot Optimization algorithm (BMEPO) for optimizing feature selection. Comparative experiments demonstrate that BMEPO exhibits stronger competitiveness in feature selection quality and classification performance compared to advanced feature selection algorithms. Finally, to further enhance the classification performance of defect prediction, a heterogeneous data stacking ensemble learning algorithm (HEDSE) based on feature selection is proposed. Experimental evaluations on 16 open-source software defect datasets indicate that the proposed HEDSE outperforms existing methods, providing a novel and effective solution for software defect prediction. The proposed approaches in this study hold significant practical value, particularly in improving software quality, optimizing testing resource allocation, and reducing maintenance costs, offering broad potential for application in real-world software engineering scenarios.

Introduction

With the rapid development of informatization, digitization, and intelligence, software has become increasingly important. Software-defined networking, software-defined storage, software-defined data centers, and the software-defined world have become hot topics. Therefore, ensuring the quality of software has become a top priority for developers and testers. In software testing, there is the Pareto principle (Persson & Nicklasson, 2022), known as the 80–20 rule, where 80% of defects exist in 20% of software modules. Rapid identification of these modules 20% defect-prone plays a crucial role in guiding the allocation of testing resources and designing test cases at later stages. Therefore, predicting and discovering defects in the early stages of software development can significantly enhance software quality by reducing debugging and maintenance costs and ensuring more efficient resource allocation throughout the development lifecycle.

Software defect prediction techniques are mainly based on extracting software defect-related metrics from historical databases. These techniques use machine learning, statistical methods, and other methods to build models to predict defects in software modules. Software defect prediction involves categorizing modules into two classes: defective and non-defective. Previous research has found connections between software module metrics and defects (Song et al., 2010). Studies have used various machine learning models, including decision trees, logistic regression, support vector machines (Husin & Pribadi, 2022; Richards, 2022), and even deep learning models such as CNN (Zain et al., 2022), to predict software defects using software module metric datasets. The use of these classification models has significantly contributed to improving the quality of the software. However, existing defect prediction techniques often use all the features of metric data without considering the potential impact of different features on the accuracy of the classification model. Identifying effective features for classification models is an NP problem, for which evolutionary algorithms can be used (Papa et al., 2018).

This article introduces an innovative method to predict software defects. Firstly, it presents a multi-strategy enhanced parrot optimization algorithm that has been rigorously tested on various functions, demonstrating its exceptional effectiveness. Next, a binary variant of the algorithm is developed to select the most influential features of the training dataset and applied to support vector machine (SVM), decision trees, k-nearest neighbors (KNN), and other classification models. Comparison with other feature selection algorithms confirms the superiority of the proposed algorithm. Finally, to further enhance the classification models’ performance, a feature selection-based heterogeneous data stacking ensemble defect prediction model is introduced.

The study presents the following significant contributions: (1) We have devised a multi-strategy enhanced Parrot Optimization (MEPO) and tested it alongside the original parrot optimization algorithm on eight test functions. The results of our experiments clearly demonstrate that the enhanced algorithm surpasses the original algorithm in both convergence speed and optimization accuracy of the optimal solution.

(2) Our application of the multi-strategy enhanced parrot optimization algorithm to feature selection in defect prediction metrics has shown its superior performance when compared to three commonly used algorithms. The binary BMEPO exhibits remarkable competitiveness in addressing feature selection.

(3) For the first time, we have introduced a heterogeneous data stacking ensemble learning model and applied it to the field of defect prediction. Through rigorous experimental comparisons, this model has been shown to outperform homogeneous data stacking ensemble learning models.

The remainder of this study is organized as follows. “Related Work” briefly describes the relevant background knowledge of the multi-strategy enhanced parrot optimizer feature selection scheme and the software defect prediction classification framework. “The Approach” elaborates on the software defect prediction framework based on the multi-source heterogeneous hybrid deep learning model. “Experimental Results and Analysis” applies the proposed method to the field of defect prediction and presents our experimental results and analysis. Finally, “Conclusion and Future Works” concludes the article and discusses plans for future work.

Related work

Recently, researchers have proposed various machine learning methods to address the problem of software defect prediction, achieving significant success (Rathore & Kumar, 2019). This section provides a detailed overview of related work in the field of software defect prediction, as well as the meta-heuristic learning algorithms and ensemble learning algorithms involved in this study.

Related work on software defect prediction

Supervised and unsupervised machine learning techniques have been employed for software defect prediction, including: SVM (Singh, Kaur & Malhotra, 2009), decision trees (DT) (Khoshgoftaar & Seliya, 2003), Bayesian networks (BN) (Yuan et al., 2000), naive Bayes (NB) (Menzies, Greenwald & Frank, 2006), KNN (Kumar Dwivedi & Singh, 2016), multilayer perceptron (MLP) (Carrozza et al., 2013), artificial neural networks (ANN) (Goyal & Bisi, 2015), logistic regression (LR) (Caglayan et al., 2015), multinomial logistic regression (MLR) (Carrozza et al., 2013), random forests (RF) (Bowes, Hall & Petrić, 2018), and unconstrained MLP (Thaher & Khamayseh, 2021).

Khoshgoftaar, Bullard & Gao (2009) are among the pioneers in developing feature selection (FS) techniques for imbalanced data sets. In their research, they applied various wrapper-based feature-ranking and feature subset selection techniques to generate candidate feature sets and combined them with different classification algorithms for prediction. The results showed that applying superior feature selection methods can enhance the performance of software defect prediction (SDP) models and make them more applicable to real-world scenarios. Arar & Ayan (2015) proposed a cost-sensitive ANN based ABC algorithm for software defect prediction (SDP). In this study, feature selection was achieved using correlation-based FS techniques in the Waikato Environment for Knowledge Analysis (WEKA) tool. The results indicated that reducing the number of features did not significantly impact prediction accuracy. Jayanthi & Florence (2019) employed a feature reduction method called principal component analysis (PCA). They addressed errors through maximum likelihood estimation, reducing the potential for errors during the construction of PCA features. The selected features were then fed into a neural network-based classification technique for error prediction. Manjula & Florence (2019) proposed a hybrid method for error prediction that combined the genetic algorithm (GA) for feature optimization and deep neural networks (DNN) for classification. Experiments on the Promise dataset for software defect prediction demonstrated improved classification accuracy with the proposed hybrid method. Xu et al. (2019) introduced a new SDP framework called KPWE, which combines kernel principal component analysis (KPCA) and weighted extreme learning machine (WELM), focusing on feature extraction. Cai et al. (2020) proposed a hybrid multi-objective cuckoo search subsampling software defect prediction model based on SVM (HMOCS-US-SVM). This model uses three undersampling methods to determine the decision domain range and employs a hybrid multi-objective cuckoo search with dynamic local search (HMOCS) to optimize SVM parameters, addressing the class imbalance in defect datasets and parameter selection issues in support vector machines.

The aforementioned methods indicate that the application of preprocessing techniques, such as feature selection, can significantly impact the performance of software defect prediction models. In conjunction with the No Free Lunch (NFL) program theorem in optimization (Wolpert & Macready, 1997), which states that no single classifier is universally optimal for all possible classification problems (Bhaskar, Hoyle & Singh, 2006), this has motivated us to propose an advanced method for predicting software defects. This method treats the stacking ensemble learning classifier as a machine learning technique while employing advanced feature selection methods as wrappers. Specifically, we utilize the novel multi-strategy enhanced parrot optimizer as a search strategy to select the optimal features.

Parrot optimization algorithm

The Parrot Optimization algorithm (POA) (Lian et al., 2024) is a swarm intelligence optimization algorithm inspired by the behaviors of foraging, roosting, communication and fear of strangers observed in domesticated parrots. In each iteration, each individual in POA randomly exhibits one of these four behaviors, accurately simulating the behavioral randomness observed in domesticated parrots. This significantly increases the diversity of the population. Furthermore, by eliminating the traditional two-phase exploration-exploitation structure, POA effectively reduces the risk of getting trapped in local optima.

In the foraging behavior, parrots estimate the approximate location of food by observing its position or considering the location of their owner, then fly towards their respective positions. The positional movement follows the following equations:

(1) Xit+1=(Xit−Xbest)×Levy(dim)+rand(0,1)×(1−tMaxiter)2tMaxiter×Xmeant

where Xit+1 represents the position to be updated, Xit denotes the current position, Xmeant signifies the mean value within the current population, Xbest represents the best position found from initialization to the current search, Levy(dim) indicates the Levy distribution, t stands for the current iteration number, and Maxiter denotes the maximum number of iterations.

The perching behavior is used to describe when a parrot suddenly flies to any part of its owner’s body and remains stationary there for some time. The perching movement follows the following equation:

(2) Xit+1=Xit+Xbest×Levy(dim)+rand(0,1)×ones(1,dim)

where ones(1, dim) represents a vector of all ones with dimension dim.

In the communication behavior, the parrot either flies towards the flock or does not fly towards the flock. In the Parrot Optimization Algorithm, the probabilities of these two behaviors are equal. The communication behavior follows the following equation:

(3) Xit+1={0.2×rand(0,1)×(1−tMaxiter)×(Xit−Xmeant),ifP≤0.50.2×rand(0,1)×exp⁡(−trand(0,1)×Maxiter),ifP > 0.5

where P≤0.5 represents flying towards the flock, P > 0.5 represents flying away from the flock.

Parrots exhibit an inherent fear of strangers, maintaining a distance from unfamiliar individuals. They engage in behaviors aimed at seeking a safe environment together with their owners. The fear behavior towards strangers can be represented by the following equation:

(4) Xit+1=Xit+rand(0,1)×cos⁡(12π×tMaxiter)×(Xbest−Xit)−cos⁡(rand(0,1)×π)×(tMaxiter)2Maxiter×(Xit−Xbest).

Stacked ensemble learning

Stacked ensemble learning (Agarwal & Chowdary, 2020) integrates the predictions of multiple models to achieve the final prediction. Its core idea is to use the outputs of various models as new features input into a meta-model, which is then used for final prediction. Stacked ensembles typically consist of two main stages: first, training multiple different machine learning models on the training data, which can be classifiers, regressors, or other types of models; second, utilizing the predictions generated by the base models on the validation set as new features and inputting them into a meta-model. The meta-model is then trained to obtain the final prediction. The framework diagram of stacked ensemble learning is illustrated in Fig. 1.

Figure 1 Stacked ensemble learning framework.

The approach

The main objective of this study is to develop precise defect classification models by using feature selection. In the data preparation phase, the defect measurement data from the Promise project software is normalized and then divided into training and testing sets. Oversampling is applied to balance the data in the training set. In the feature selection phase, a multi-strategy improved Parrot Optimizer feature selection scheme is proposed for the measurement data. The optimal measurement features for SVM, KNN, and decision tree classification models are obtained through meta-heuristic search. Finally, to further improve the performance of the defect classification models, a heterogeneous data stacking ensemble learning classification model is created based on the optimal features of each model. The overall architecture is illustrated in Fig. 2.

Figure 2 Overall architecture.

Dataset acquisition and preprocessing

Traditional static code quality metrics have been proven to be related to software defects (Koru & Liu, 2007; El Emam et al., 2001; Nagappan, Ball & Zeller, 2006; Fenton & Ohlsson, 2000). In this study, 21 defect datasets were selected from the Promise repository, each containing 20 software measurement metrics. The measurement metrics are shown in Table 1, and the number of files and defect file probability in each dataset are shown in Table 2. The Promise dataset is widely used in software defect prediction research because of its origin in real-world software projects, providing significant practical relevance and broad applicability. The datasets not only reflect project characteristics under various development environments but also consider multiple software metrics, including code complexity, comment density, and functional defects, all of which are closely related to the occurrence of software defects. Due to the different dimensions of each software measurement metric, to avoid bias towards certain key features, standardization processing is performed on the metadata of each measurement. The standardization formula is as follows:

(5) xi~=xi−x¯δ

where x¯ is the mean of the dataset, δ is the standard deviation of the dataset.

Table 1 Software quality metrics.

0. Weighted Methods per Class (wmc)	1. Depth of Inheritance Tree (dit)	
2. Number of Children (noc)	3. Coupling Between Objects (cbo)	
4. Response for Class (rfc)	5. Lack of Cohesion in Methods (lcom)	
6. Abstract Classes (ca)	7. Efferent Coupling (ce)	
8. Number of Public Methods (npm)	9. Lack of Cohesion in Methods (lcom3)	
10. Lines of Code (loc)	11. Data Abstraction Coupling (dam)	
12. Method Abstraction Coupling (moa)	13. Method Functionality Abstraction (mfa)	
14. Class Abstractness (cam)	15. Afferent Coupling (ic)	
16. Number of Base Classes (cbm)	17. Average Method Complexity (amc)	
18. Maximum Cyclomatic Complexity (max_cc)	19. Average Cyclomatic Complexity (avg_cc)	

Table 2 Defect dataset.

Dataset	No. of files	Num. of defect-free	Num. of defective	Defect file (%)	
ant1.7	745	579	166	22.28%	
camel1.0	339	326	13	3.83%	
camel1.2	608	392	216	35.53%	
camel1.4	872	727	145	16.63%	
camel1.6	965	777	188	19.48%	
jedit3.2	272	182	90	33.09%	
jedit4.0	306	231	75	24.51%	
jedit4.1	312	233	79	25.32%	
jedit4.2	367	319	48	13.08%	
jedit4.3	492	481	11	2.24%	
log4j1.0	135	101	34	25.19%	
log4j1.1	109	72	37	33.94%	
poi2.0	314	277	37	11.78%	
poi2.5	385	137	248	64.42%	
poi3.0	442	161	281	63.57%	
synapse1.2	256	170	86	33.59%	
velocity1.6	229	151	78	34.06%	
xalan2.4	723	613	110	15.21%	
xerces1.2	440	369	71	16.14%	
xerces1.3	453	384	69	15.23%	
xerces1.4	588	151	437	74.32%	

By analyzing the defect dataset in Table 2, it was found that the number of defective files in most datasets is relatively small, resulting in a highly imbalanced state of data categories. To balance the dataset, this study employs the SMOTE oversampling technique.

Multi-strategy improved parrot optimizer

In response to the dependency on initial population, premature convergence, and poor global search capability inherent in the Parrot Optimization Algorithm, an improved Parrot Optimization algorithm is proposed. This algorithm first utilizes the Tent chaotic mapping to generate initial positions of parrot populations, laying the foundation for population diversity during the global search process. Secondly, an adaptive t-distribution mutation method is employed to perturb individual positions, enhancing the algorithm’s ability to escape local optima. Finally, a winner-take-all biological competition elimination strategy is introduced, where the parrot populations are mutated by the elite reverse learning strategy, and the original populations are eliminated based on fitness values, thereby strengthening the algorithm’s development capability.

Tent chaotic mapping

During the initialization stage, the POA uses a random number generation method to create the initial population. However, this method leads to uneven distribution across the solution space, which hinders population diversity. To address this, the algorithm employs chaotic mapping, known for its randomness and ability to cover the entire search space globally. This can help speed up convergence and prevent the algorithm from getting stuck in local optima (Arora, Sharma & Anand, 2020). In this study, the Parrot population is initialized using the Tent chaotic mapping method, expressed mathematically as follows:

(6) xn+1={xnu,if0≤xn≤u1−xn1−u,ifu≤xn≤1

where xn represents the current state of the chaotic system, xn+1 represents the next state calculated based on the current state xn, u is the chaotic state parameter, and this article adopts a value of 0.5 for u. The parameter value 0.5 typically lies within the stable region or at the transition point of chaotic systems, ensuring that the generated sequence exhibits chaotic characteristics without being entirely random. In numerical simulations, this value neither leads to complete system instability nor does it restrict the diversity of the generated sequence, providing an adequate balance between chaos and controllability. The population distribution and histogram generated by the Tent map, as derived from Eq. (6), are shown in Figs. 3 and 4, respectively. As shown in Figs. 3 and 4, the initial population distribution generated by the Tent map exhibits good randomness and is relatively uniform.

Figure 3 Tent map chaos population distribution.

Figure 4 Tent map chaos histogram distribution.

T-distribution mutation

After completing the update step of the parrot individuals, to further enhance the optimization capability of the POA, this article introduces a mutation strategy based on the t-distribution to generate new individual positions. The t-distribution mutation operator combines the advantages of the Gaussian operator and the Cauchy operator, which helps the algorithm search for better solutions in a wider search space (Li et al., 2020; Mostafa et al., 2020). The individual position update equation for t-distribution mutation is as follows:

(7) xnew=xi+t(iter)×xi

where xi represents the position of the i-th individual, xnew represents the new individual generated based on the i-th individual’s position, and t(iter) follows a t-distribution with degrees of freedom equal to the number of iterations.

When the algorithm starts, the number of iterations is small, the mutation effect of the t-distribution is similar to that of the Cauchy distribution. Figure 5 illustrates the scatter plot of the Cauchy distribution within the range of [−100, 100]. As observed, the Cauchy distribution tends to generate random numbers that are farther from the origin, causing larger disturbances in the mutation term. This results in a stronger global search capability, which helps the algorithm escape from local optima. During the mid-stage of the algorithm, the t-distribution mutation lies between the Cauchy and Gaussian distributions. In the later stages, with a larger number of iterations, the mutation behavior of the t-distribution becomes equivalent to that of the Gaussian distribution. Figure 6 shows the scatter plot of the Gaussian distribution. The random numbers generated by the Gaussian distribution are closer to the origin, resulting in smaller perturbations from the mutation term. This leads to a more detailed local search, improving the algorithm’s convergence accuracy and enhancing its local exploitation ability.

Figure 5 Scatter plot of the Cauchy distribution.

Figure 6 Scatter plot of the Gaussian distribution.

After generating new individual positions, this article follows a greedy strategy to comprehensively evaluate the newly generated individuals with the existing individuals, selecting the optimal individual as the solution for the current step. This strategy ensures that the individuals are updated in the direction of better solutions, thereby improving the optimization capability of the algorithm.

Elite reverse learning strategy

The reverse learning strategy was proposed by Tizhoosh (2005), which can increase the diversity and quality of the population. Tizhoosh’s research indicates that the probability of reverse solutions approaching the global optimum is 50% higher than that of current solutions. The formula for the reverse strategy is as follows:

(8) x~ij=w×(lj+uj)−xij

where w represents the coefficient following a uniform distribution on [0,1], xij denotes the j-dimensional solution of the i-th population, x~ij represents the reverse solution in the j-th dimension of the i-th population, lj stands for the minimum value of all populations in the j-th dimension, and uj denotes the maximum value of all populations in the j-th dimension.

The introduction of reverse solution strategies effectively expands the search range of the algorithm, providing potential pathways for the algorithm to escape local optima. In each population update process, we adopt an elite retention mechanism, sorting and evaluating the current solutions and their corresponding reverse solutions based on the fitness function, and then selecting the best-performing individuals as elite individuals in the next-generation population. This approach aims to maintain the excellent genes of the population while promoting the algorithm’s global search capability, in pursuit of more ideal solutions.

Multi-strategy enhanced parrot optimization algorithm procedure

The specific procedure of the multi-strategy enhanced Parrot Optimization algorithm is as follows: Step 1: Initialize the parrot population using Tent chaotic mapping as described in Eq. (6) and then compute the fitness.

Step 2: Employ the reverse learning strategy to generate reverse solutions according to Eq. (8). Compare the original population with the reverse population, and select populations with higher fitness to join the next generation of parrot populations.

Step 3: For each individual, randomly select an updating strategy. If it is foraging behavior, update the individual’s position according to Eq. (1). If it is resting behavior, update the individual’s position according to Eq. (2). If it is social behavior, update the individual’s position according to Eq. (3). If it is fear behavior, update the individual’s position according to Eq. (4).

Step 4: Optimize the parrot’s positions through t-distribution mutation according to Eq. (7). Compare the fitness values before and after optimization, and select the optimal individuals from the original and new positions using a greedy strategy.

Step 5: Determine whether the maximum number of iterations has been reached. If the maximum number of iterations has been reached, output the optimal parrot positions and fitness values. Otherwise, return to Step 2 and continue iterating.

The flowchart of the proposed MEPO is illustrated in Fig. 7. The pseudo-code of MEPO is shown in Table 3.

Figure 7 Flowchart of MEPO algorithm.

Table 3 Pseudo-code of MEPO.

Algorithm 1 MEPO algorithm	
 1: Initialize the MEPO parameters	
 2: Initialization of the Parrot Population Using the Tent Chaotic Map	
 3: for i=1 to Max_iter do	
 4:    Generate the elite population based on Eq. (8) and the greedy algorithm.	
 5:    Calculate the fitness function	
 6:    Find the best position	
 7:    for j=1 to N do	
 8:       St=np.random.randint(1,5)	
 9:      if St=1 then	
10:        Behavior 1: The foraging behavior	
11:        Update position by Eq. (1)	
12:      else if St=2 then	
13:        Behavior 2: The perching behavior	
14:        Update position by Eq. (2)	
15:      else if St=3 then	
16:        Behavior 3: The communicating behavior	
17:        Update position by Eq. (3)	
18:      else if St=4 then	
19:        Behavior 4: The fear of strangers’ behavior	
20:        Update position by Eq. (4)	
21:      end if	
22:    end for	
23:    Update position based on Eq. (6) and the greedy algorithm.	
24: end for	
25: Return the best position	

Feature selection of software metrics using binary multi-strategy enhanced parrot optimization algorithm

The process of selecting the most relevant metrics is a complex task that involves optimizing multiple objectives. In this particular study, we are focusing on classifying software defects, and we are using the AUC metric to evaluate the performance of our classification models. The AUC (area under the ROC curve) is a crucial measure, and a higher value indicates better classification performance. For this study, we have chosen AUC as the fitness function. When developing solutions, we need to consider how these solutions are represented. In our case, solutions are represented as binary vectors with N elements, where N is the total number of metrics in the original dataset. Each element in the vector is a binary value: 0 means the corresponding metric is not selected, and 1 means it is selected. Figure 8 illustrates this feature selection scheme. The Parrot Optimization (PO) method has shown significant advantages in solving continuous optimization problems. In this study, we have integrated the MEPO algorithm, which uses binary encoding, to develop the BMEPO algorithm. While MEPO is designed for continuous optimization problems, BMEPO is specifically developed to handle binary optimization problems. This approach effectively resolves the optimization problem of metric feature selection.

Figure 8 Binary solution representation.

Heterogeneous data stack ensemble learning framework

Traditional stacking methods often assume that all base models should be trained on the same dataset, ignoring the differences in data feature selection among different models. This study proposes a heterogeneous data stack ensemble framework for software defect prediction. Firstly, for the defect dataset, optimal features adapted to each model are extracted using BMEPO. Secondly, primary learners are trained on feature-filtered data based on different feature selectors, and the outputs serve as inputs to a secondary learner, the logistic regression model. After training the logistic regression model, final predictions are made. The overall framework is illustrated in Fig. 9.

Figure 9 Heterogeneous data stack ensemble learning framework.

Experimental results and analysis

In this section, we first conducted performance tests on the MEPO to verify its effectiveness. Subsequently, we applied the multi-strategy enhanced parrot optimization algorithm to the field of feature selection for software defect prediction and conducted comparative experiments with its binary-improved version. The experimental results demonstrated that the BMEPO achieved better performance in feature selection. Furthermore, to further explore the potential of the selected features in software defect prediction, we conducted comparative experiments using heterogeneous data stack ensemble learning (HEDSE), comparing it with the same data stack ensemble and single classification models. The superiority of HEDSE was confirmed through these comparisons.

The experiments were implemented using Python 3.8.3 on a PC with an Intel Core (TM) i7-8550U 2.10 GHz CPU and 8 GB RAM, running Windows 10. The proposed classification framework was implemented with Python, utilizing open-source libraries (e.g., Panda, Numpy, Matplotlib, and SKlearn (Scikit-learn)).

Result analysis of MEPO and experimental comparison with other algorithms

Algorithm comparison on test functions

To demonstrate the superiority of the multi-strategy enhanced parrot optimization algorithm, this study selected several test functions with different characteristics for optimization and compared them with the Parrot Optimization algorithm. To ensure the fairness of the experiments, the number of parrots in all experiments was set to 30, and the maximum number of iterations was set to 500. To avoid result bias caused by randomness, the algorithm was independently run 50 times for each function. Table 4 lists the basic information of the test functions, and Table 5 presents the experimental results of MEPO and PO after 50 independent runs on multiple standard test functions.

Table 4 Test functions and specific information.

Function name	Function	Dimension	Range	Best solution	
Rosenbrock’s function	f1=∑i=1n−1[100(xi+1−xi2)2+(xi−1)2]	30	[−30,30]	0	
Penalized function	f2=πn{10sin⁡(πy1)+∑i=1n−1[(yi−1)2(1+10sin2(πyi+1))+(yn−1)2]}+u(xi,10,100,4)yi=1+xi+14 u(xi,a,k,m)={k(xi−a)m,if  xi > a0,if  −a < xi < ak(−xi−a)m,if  xi < −a	30	[−50,50]	0	
Penalized function	f3=πn{10sin⁡(πy1)+∑i=1n−1[(yi−1)2(1+10sin2(πyi+1))+(yn−1)2]}+u(xi,5,100,4)	30	[−50,50]	0	
Schwefel’Problem 2.26	f4=∑i=1n−xisin⁡(|xi|)	30	[−500,500]	418.98×n	
Shekel’s foxholes function	f5=(1500+∑j=1251j+∑i=12(xi−aij)6)−1	2	[−65,65]	1	
Branin function	f6=(x2−5.14π2x12+5πx1−6)2+10(1−18π)cos⁡x1+10	2	[−5,15]	0.3978	
Hartmann	f7=∑i=14ciexp⁡(−∑j=13aij(xj−pij)2)	3	[0,1]	−3.8627	
Hartmann	f8=∑i=14ciexp⁡(−∑j=13aij(xj−pij)2)	3	[0,1]	−3.3223	

Table 5 Results of test functions (different algorithms).

Statistic	Algorithms	f1	f2	f3	f4	f5	f6	f7	f8	
Best	MEPO	3.4E−15	4.31E−20	1.7E−26	12,569.486	0.998	0.3978	−3.863	−3.322	
	PO	4.19E−13	2.79E−9	6.25E−14	−11,483.239	0.998	0.398	−3.861	−3.169	
Mean	MEPO	0.006	0.016	2.75E−4	−9,772.386	4.629	0.398	−3.862	−3.276	
	PO	0.015	0.023	3.93E−4	−7,453.684	7.974	0.404	−3.833	−2.814	
Worst	MEPO	0.149	0.093	0.012	−6,764.510	12.670	0.402	−3.861	−3.086	
	PO	0.162	0.105	0.016	−3,453.645	12.670	0.5111	−3.739	−2.281	
STD	MEPO	0.022	0.023	0.002	1,634.980	4.838	6.74E−4	4.09E-4	0.068	
	PO	0.031	0.093	0.012	1,971.884	5.272	0.017	0.027	0.229	
Time (s)	MEPO	1.684	3.656	3.124	1.565	8.066	1.477	3.306	3.189	
	PO	1.204	2.401	2.023	1.112	7.411	1.025	1.905	1.982	

Upon analyzing Table 5, it is evident that the MEPO and PO algorithms both reach the optimal value for the f5 and f6 test functions. However, for the other test functions, the MEPO algorithm consistently obtains lower optimal values than the PO algorithm. Additionally, when comparing the mean values, worst values, and variances, it is clear that the MEPO algorithm consistently outperforms the PO algorithm. This indicates that the MEPO algorithm possesses superior optimization capabilities compared to the PO algorithm.

To further verify the convergence of the MEPO algorithm, iterative convergence curves for test functions were plotted, with the horizontal axis representing the number of iterations and the vertical axis representing the average value over 50 independent runs. As shown in the figure below, on the f6 test function, the MEPO and PO algorithms converge almost simultaneously. In the f2, f5 and f8 test functions, the MEPO algorithm converges later than the PO algorithm, but the average value at convergence is lower than that of the PO algorithm. In the f1, f3, f4 and f7 test functions, the MEPO algorithm converges earlier than the PO algorithm, and the average value at convergence is lower than that of the PO algorithm. The iterative convergence curves for the test functions are shown in Fig. 10.

Figure 10 Comparison of convergence for MEPO and PO algorithms.

In conclusion, the MEPO algorithm, due to the incorporation of mechanisms such as chaotic mapping, reverse solution generation, and t-distribution mutation, incurs a slight disadvantage in terms of computation time. However, when applied to optimization problems, it demonstrates better performance and convergence compared to the PO algorithm.

Parameter sensitivity analysis

In the comparative experiments mentioned above, the default behavior proportions in MEPO foraging (F), resting (S), communication (C) and fear (A) were set to 1:1:1:1. However, it is hypothesized that varying these behavioral distributions may yield different optimization outcomes. A parameter sensitivity analysis experiment was designed to explore the impact of different behavioral distributions on optimization performance.

Five variants of the MEPO algorithm with different behavioral proportions were proposed: the original MEPO algorithm with F:S:C:A = 1:1:1:1, the MEPO-F algorithm with F:S:C:A = 2:1:1:1, the MEPO-S algorithm with F:S:C:A = 1:2:1:1, the MEPO-C algorithm with F:S:C:A = 1:1:2:1, and the MEPO-A algorithm with F:S:C:A = 1:1:1:2. Under the condition of maintaining the same size and number of iterations of the parrot population, only the behavioral distribution ratios were altered. The optimal solutions for these algorithm variants were calculated in eight test functions. The results of the comparison experiments are presented in Table 6.

Table 6 Mean results of test functions (different parameter).

Algorithm	f1	f2	f3	f4	f5	f6	f7	f8	
MEPO	0.006	0.016	2.75E−4	−9,772.386	4.629	0.398	−3.862	−3.276	
MEPO-F	0.164	0.018	4.97E−5	−9,468.191	8.751	0.398	−3.861	− 3.256	
MEPO-S	0.281	0.019	0.001	−9,643.177	8.582	0.398	−3.861	−3.263	
MEPO-C	0.205	0.030	1.19E-4	−9,635.884	8.403	0.398	−3.860	−3.276	
MEPO-A	0.471	0.022	3.45E-4	−10,230.171	7.633	0.398	−3.861	−3.237	
Note:

Best values for each task are presented in bold.

Based on the experimental results, the MEPO algorithm demonstrates superior optimization performance in several test functions. Specifically, MEPO achieved the best solution in the functions f1, f2, f5, and f7, indicating that a balanced distribution of behaviors effectively coordinates global exploration and local exploitation, leading to optimal results in these problems. In contrast, for the f6 function, all algorithms yielded identical results, suggesting that the problem may not exhibit sensitivity to variations in the behavioral distribution, or that all algorithms converge to similar solutions for this particular problem.

In the case of the f3 function, the MEPO-F variant, which emphasizes the foraging behavior, obtained the best result. This indicates that for problems with a higher demand for local exploitation, strengthening the foraging behavior enhances the algorithm’s ability to search the local solution space more effectively. In the f4 and f8 functions, the MEPO-A variant, which emphasizes fear behavior, performed best, suggesting that for problems with a stronger requirement for global exploration, fear behavior plays a critical role in promoting search diversity and preventing premature convergence to local optima.

In summary, while adjusting the behavioral distribution can further enhance the algorithm’s performance on certain problems—such as MEPO-F excelling on the f3 function and MEPO-A outperforming other variants on the f4 and f8 functions—MEPO, with its default behavior distribution, remains highly effective and well-balanced for most optimization tasks. These findings underscore the adaptability of MEPO to different problem characteristics and suggest that tailored adjustments to behavior distribution could be a promising avenue for further optimization.

Result analysis of BMEPO and experimental comparison with other algorithms

In the software defect datasets, we use the Whale Optimization Algorithm (Nasiri & Khiyabani, 2018) Feature Selector (BWO), Parrot Optimization Algorithm Feature Selector (BPO), and Crowned Porcupine Optimization Algorithm (Abdel-Basset, Mohamed & Abouhawwash, 2024) Feature Selector (BCPO) as baseline algorithms to compare with the Binary Multi-strategy Enhanced Parrot Optimization Algorithm Feature Selector (BMEPO). The population size is set to 10, the maximum number of iterations to 50, and the dimensionality of the population of individuals to 20. These four feature selection algorithms are applied to 20 defect datasets from the Promise repository. The classification models selected are decision tree, SVM, and KNN, with default parameters for each machine learning model. The Area Under the Curve (AUC) metric is used to evaluate the performance of the feature selection algorithms. Each algorithm is executed 30 times to ensure fairness, and we report the average and variance of the AUC values over these 30 runs. The results of the comparison experiments are presented in Tables 7–9, and Fig. 11. To further evaluate the computational efficiency of the algorithms, we conduct a comparative experiment on the average runtime of the BMEPO, BPO, BWO, and BCPO algorithms across 16 datasets using DT, SVM, and KNN models. The experimental results are shown in Table 10.

Table 7 Feature selection comparison results based on AUC values using the decision tree mode.

Data set	Measure	BMEPO	BPO	BWO	BCPO	
Ant-1.7	AVG	0.7593	0.7488	0.7181	0.7465	
	STD	0.0079	0.0123	0.0166	0.0116	
Camel-1.0	AVG	0.8797	0.8532	0.7383	0.8365	
	STD	0.0440	0.0054	0.0789	0.0402	
Camel-1.2	AVG	0.6964	0.6851	0.6657	0.6906	
	STD	0.0141	0.0164	0.0221	0.0103	
Jedit-4.0	AVG	0.7355	0.7176	0.6789	0.7212	
	STD	0.0145	0.0189	0.0297	0.0139	
Jedit-4.1	AVG	0.8105	0.7948	0.7723	0.8001	
	STD	0.0101	0.0169	0.0204	0.0151	
Jedit-4.2	AVG	0.8793	0.8672	0.8239	0.8660	
	STD	0.0138	0.0123	0.0290	0.0171	
Jedit-4.3	AVG	0.9705	0.9081	0.7741	0.9306	
	STD	0.0071	0.0659	0.0733	0.0476	
Log4j-1.0	AVG	0.8207	0.7929	0.7722	0.8118	
	STD	0.0127	0.0110	0.0212	0.0130	
Log4j-1.1	AVG	0.8503	0.8198	0.7698	0.8402	
	STD	0.0086	0.0294	0.0410	0.0221	
Poi-2.0	AVG	0.7960	0.7708	0.7249	0.7781	
	STD	0.0198	0.0242	0.0297	0.0247	
Poi-2.5	AVG	0.8676	0.8467	0.8211	0.8484	
	STD	0.0121	0.0150	0.0228	0.0143	
Poi-3.0	AVG	0.8540	0.8404	0.8234	0.8460	
	STD	0.0113	0.0074	0.0107	0.0103	
Synapse-1.2	AVG	0.8213	0.7917	0.7544	0.7963	
	STD	0.0166	0.0226	0.0212	0.0175	
Velocity-1.6	AVG	0.8170	0.7850	0.7635	0.7857	
	STD	0.0164	0.0294	0.0293	0.0215	
Xalan-2.4	AVG	0.7731	0.7504	0.7260	0.7609	
	STD	0.0154	0.0156	0.0284	0.0157	
Xerces-1.3	AVG	0.8186	0.7949	0.7328	0.8002	
	STD	0.0114	0.0318	0.0235	0.0224	

Table 8 Feature selection comparison results based on AUC values using the SVM mode.

Data set	Measure	BMEPO	BPO	BWO	BCPO	
Ant-1.7	AVG	0.8152	0.8050	0.7955	0.8080	
	STD	0.0075	0.0083	0.0122	0.0082	
Camel-1.0	AVG	0.6964	0.6851	0.6657	0.6906	
	STD	0.0153	0.0208	0.0444	0.0174	
Camel-1.2	AVG	0.6809	0.6690	0.6459	0.6722	
	STD	0.0106	0.0113	0.0159	0.0113	
Jedit-4.0	AVG	0.7185	0.7121	0.6811	0.7094	
	STD	0.0053	0.0082	0.0165	0.0079	
Jedit-4.1	AVG	0.8441	0.8260	0.8035	0.8300	
	STD	0.0084	0.0155	0.0228	0.0140	
Jedit-4.2	AVG	0.9129	0.9039	0.8813	0.9058	
	STD	0.0036	0.0085	0.0225	0.0066	
Jedit-4.3	AVG	0.8845	0.8334	0.7253	0.8335	
	STD	0.0258	0.0315	0.0809	0.0389	
Log4j-1.0	AVG	0.8979	0.8703	0.8202	0.8793	
	STD	0.0157	0.0236	0.0309	0.0200	
Log4j-1.1	AVG	0.8200	0.7982	0.7734	0.8083	
	STD	0.0090	0.0127	0.0234	0.0121	
Poi-2.0	AVG	0.8575	0.8219	0.7795	0.8350	
	STD	0.0126	0.0265	0.0336	0.0222	
Poi-2.5	AVG	0.8522	0.8449	0.8298	0.8471	
	STD	0.0050	0.0071	0.0125	0.0059	
Poi-3.0	AVG	0.8515	0.8430	0.8113	0.8439	
	STD	0.0053	0.0071	0.0094	0.0091	
Synapse-1.2	AVG	0.8076	0.7925	0.7748	0.7955	
	STD	0.0093	0.0119	0.0190	0.0104	
Velocity-1.6	AVG	0.7733	0.7574	0.7346	0.7631	
	STD	0.0069	0.0126	0.0209	0.0092	
Xalan-2.4	AVG	0.7805	0.7688	0.7370	0.7643	
	STD	0.0058	0.0096	0.0206	0.0097	
Xerces-1.3	AVG	0.8355	0.8192	0.7932	0.8200	
	STD	0.0113	0.0148	0.0171	0.0115	

Table 9 Feature selection comparison results based on AUC values using the KNN mode.

Data set	Measure	BMEPO	BPO	BWO	BCPO	
Ant-1.7	AVG	0.8012	0.7851	0.7694	0.7916	
	STD	0.0105	0.0119	0.0140	0.0122	
Camel-1.0	AVG	0.9444	0.9326	0.8727	0.9369	
	STD	0.0065	0.0122	0.0465	0.0099	
Camel-1.2	AVG	0.6745	0.6598	0.6445	0.6680	
	STD	0.0082	0.0097	0.0147	0.0121	
Jedit-4.0	AVG	0.7616	0.7482	0.7167	0.7513	
	STD	0.0098	0.0126	0.0172	0.0134	
Jedit-4.1	AVG	0.8537	0.8280	0.8036	0.8418	
	STD	0.0096	0.0162	0.0149	0.0166	
Jedit-4.2	AVG	0.8988	0.8766	0.8252	0.8794	
	STD	0.0155	0.0142	0.0369	0.0186	
Jedit-4.3	AVG	0.9535	0.9355	0.8284	0.9379	
	STD	0.0081	0.0250	0.0513	0.0151	
Log4j-1.0	AVG	0.8741	0.8569	0.7989	0.8558	
	STD	0.0113	0.0126	0.0303	0.0168	
Log4j-1.1	AVG	0.8928	0.8376	0.7996	0.8533	
	STD	0.0262	0.0254	0.0403	0.0281	
Poi-2.0	AVG	0.8240	0.8061	0.7725	0.8101	
	STD	0.0142	0.0152	0.0209	0.0121	
Poi-2.5	AVG	0.8624	0.8562	0.8385	0.8578	
	STD	0.0055	0.0113	0.0190	0.0074	
Poi-3.0	AVG	0.8577	0.8386	0.8182	0.8442	
	STD	0.0076	0.0100	0.0213	0.0124	
Synapse-1.2	AVG	0.8137	0.8012	0.7851	0.8082	
	STD	0.0123	0.0129	0.0176	0.0124	
Velocity-1.6	AVG	0.8068	0.7836	0.7601	0.7932	
	STD	0.0081	0.0165	0.0264	0.0183	
Xalan-2.4	AVG	0.8049	0.7850	0.7647	0.7946	
	STD	0.0094	0.0100	0.0198	0.0127	
Xerces-1.3	AVG	0.8317	0.8022	0.7732	0.8100	
	STD	0.0209	0.0301	0.0356	0.0215	

Figure 11 AUC plot For DT/SVM/KNN models.

Table 10 Average runtime (s).

Algorithm	DT	SVM	KNN	
BMEPO	2.606	7.417	6.544	
BPO	0.777	3.856	2.573	
BWPO	0.946	4.196	2.519	
BCPO	5.016	12.071	10.025	

As shown in Fig. 11, the feature selection performance of the BMEPO algorithm is superior to that of the BPO, BWO, and BCPO algorithms. By averaging the AUC metrics of all data in Table 7, it can be concluded that the average AUC obtained by the BMEPO algorithm for feature selection based on the Decision Tree model is 2.3 percent higher than that of the BPO algorithm, 6.8 percent higher than that of the BWO algorithm, and 1.8 percent higher than that of the BCPO algorithm. By averaging the AUC metrics of all data in Table 8, it can be concluded that the average AUC obtained by the BMEPO algorithm for feature selection based on the SVM model is 1.7 percentage points higher than that of the BPO algorithm, 5.2 percentage points higher than that of the BWO algorithm, and 1.4 percentage points higher than that of the BCPO algorithm. By averaging the AUC metrics of all data in Table 9, it can be concluded that the average AUC obtained by the BMEPO algorithm for feature selection based on the KNN model is 2 percentage points higher than that of the BPO algorithm, 5.5 percentage points higher than that of the BWO algorithm, and 1.4 percentage points higher than that of the BCPO algorithm.

As shown in Table 10, the execution time of the BMEPO algorithm is higher than that of the BPO and BWO algorithms but lower than that of the BCPO algorithm across different classification models. This difference in execution time is primarily because the BMEPO algorithm incorporates a hybrid strategy to enhance its global search capability, which increases the computational cost per iteration.

In summary, although the execution time of the BMEPO algorithm is higher compared to the BPO and BWO algorithms, its significant improvement in feature selection performance demonstrates its superiority in balancing search efficiency and feature selection quality. Moreover, unlike other algorithms that have limitations in optimization performance on specific classification models, the BMEPO algorithm outperforms the BPO, BWO, and BCPO algorithms across all classification models (decision tree, SVM, and KNN). This further validates the excellent generalization capability and stability of the BMEPO algorithm, indicating its broad applicability in feature selection tasks.

Result analysis of HEDSE and experimental comparison with other algorithms

In this section, we will discuss the impact of heterogeneous data stacking ensembles on the performance of prediction models.

First, we use the BMEPO algorithm to perform feature selection on the defect test datasets for the DT model (BMEPO-FS-DT), SVM model (BMEPO-FS-SVM), and KNN model (BMEPO-FS-KNN), respectively. We select the feature indices that yield the best values of fitness function. The selection results are shown in Table 11.

Table 11 Optimal features corresponding to different classification models.

Data set	BMEPO-FS-DT	BMEPO-FS-SVM	BMEPO-FS-KNN	
Ant-1.7	[6,8,10,11,14]	[1,4,11,15,16]	[1,4,9,10,12,14]	
Camel-1.0	[2,6,12,15,18]	[12,15,18,19]	[1,6,9,15,18,19]	
Camel-1.2	[2,3,8,11,12,13,14,16, 18,19]	[0,4,5,6,7,9,12,13,14,19]	[3,4,5,7,8,10,12,13,14,18]	
Jedit-4.0	[0,1,2,5,6,9,11,13,18]	[9,12,13,14]	[2,4,6,8,14,18]	
Jedit-4.1	[4,6,12,13,15,16,17,19]	[0,4,5,8,11,13,15,17,18,19]	[4,11,13,15,17,18]	
Jedit-4.2	[2,5,6,8,16,18,19]	[2,6,19]	[0,2,3,8,9,10,18]	
Jedit-4.3	[7,11,12,17]	[2,3,6,18]	[1,3,4,6,7,10,14,17]	
Log4j-1.0	[0,1,4,6,14,18]	[1,3,6,8,9,11]	[0,1,8,17]	
Log4j-1.1	[0,5,9,11,13,14,17,19]	[4,14,18]	[4,8,13,15,17]	
Poi-2.0	[8,10,17,18]	[0,10,11,12,15,19]	[1,2,6,11,12,16]	
Poi-2.5	[1,2,11,13,15,16]	[2,6,9,11]	[1,2,3,5,6,7,8,9,10, 11,12,13,14,15,18,19]	
Poi-3.0	[0,1,2,3,6,7,10,13, 14,15,16,17,19]	[5,6,7,9,17]	[0,3,5,9,10,14]	
Synapse-1.2	[0,2,6,8,13,18]	[0,1,4,9,11,12,14]	[7,8,10,11,16,18,19]	
Velocity-1.6	[1,7,10,11,15]	[7,10,11,17]	[2,3,4,7,10,11,12,14, 16,17,19]	
Xalan-2.4	[0,2,4,13,14,17,18]	[5,10,15,16,17]	[1,2,3,4,5,6,8,9,10,12, 13,14,19]	
Xerces-1.3	[4,13,15,19]	[0,1,3,6,8,9,13,16,18,19]	[2,3,6,7,8,10,15,17,18,19]	
Note:

*The feature index corresponds to the values in Table 1.

Second, we constructed a defect prediction model based on stacked ensemble learning, where DT, SVM, KNN, and linear regression models were employed as base learners, with a linear regression model serving as the meta-learner. The defect data underwent feature selection using the DT model feature filter, SVM model feature filter, KNN model feature filter, and without feature filter, respectively, and were then input into the stacked ensemble model for classification. For the sake of consistency in expression, we collectively refer to the aforementioned methods as the homogeneous data-stacked ensemble learning model (HODSE).

Subsequently, we utilized feature filters based on DT, SVM, and KNN models to perform feature selection on defect data, and the selected features were input into the corresponding base learners of the stacked ensemble model for classification. The results were compared with those of the homogeneous data stacked ensemble model to validate the effectiveness of the heterogeneous data stacked ensemble model. Furthermore, to comprehensively evaluate the impact of different feature selection combinations on classification performance, we designed and conducted ablation experiments to systematically analyze the performance of the DT+SVM, DT+KNN, and SVM+KNN feature filters. The experimental results are shown in Fig. 12 and Table 12.

Figure 12 Defect classification: homogeneous vs. heterogeneous models.

Table 12 Homogeneous vs. heterogeneous stacked ensemble models for defect classification.

Data set	HODSE-NOFS	HODSE-FSDT	HODSE-FSSVM	HODSE-FSKNN	HEDSE-DT+SVM	HEDSE-DT+KNN	HEDSE-SVM+KNN	HEDSE	
Ant-1.7	0.6500	0.78458	0.6987	0.7617	0.7311	0.7854	0.6875	0.8389	
Camel-1.0	0.5841	0.96938	0.6198	0.7346	0.6046	0.8546	0.6097	0.9795	
Camel-1.2	0.6350	0.66336	0.5297	0.6066	0.6317	0.7297	0.6326	0.6899	
Jedit-4.0	0.5435	0.78603	0.6216	0.6756	0.6824	0.7379	0.4962	0.7590	
Jedit-4.1	0.7070	0.7769	0.6658	0.7106	0.8034	0.8626	0.707	0.8554	
Jedit-4.2	0.5946	0.8459	0.7512	0.8510	0.6338	0.9431	0.6994	0.8926	
Jedit-4.3	0.5173	0.8195	0.5873	0.5128	0.4758	0.4862	0.4793	0.9827	
Log4j-1.0	0.6887	0.8193	0.7709	0.7032	0.7532	0.8854	0.9354	0.9016	
Log4j-1.1	0.6130	0.8511	0.8452	0.7380	0.9583	0.7797	0.5773	0.9583	
Poi-2.0	0.6758	0.7992	0.6575	0.6031	0.7467	0.848	0.6758	0.8724	
Poi-2.5	0.8197	0.8769	0.8467	0.8822	0.8644	0.8894	0.8894	0.9151	
Poi-3.0	0.7404	0.8077	0.8365	0.8143	0.7513	0.7822	0.7078	0.8702	
Synapse-1.2	0.5381	0.8607	0.7912	0.7731	0.8348	0.8602	0.74	0.8520	
Velocity-1.6	0.6344	0.74516	0.6769	0.7132	0.6373	0.8147	0.6982	0.8573	
Xalan-2.4	0.6906	0.7399	0.6244	0.6709	0.6906	0.818	0.7133	0.7822	
Xerces-1.3	0.6030	0.8783	0.6635	0.6250	0.6441	0.8747	0.5674	0.8747	
Average	0.6396	0.8140	0.6992	0.7109	0.7141	0.8111	0.6752	0.8676	
Win/Draw/Lost	0/0/16	3/0/13	0/0/16	0/0/16	0/1/15	4/0/12	1/0/15	7/1/8	
Cliff’s Delta	0.9531	0.4453	0.8593	0.7968	0.7539	0.3671	0.7734	–	
Note:

Best values for each task are presented in bold.

From Table 12 and Fig. 12, it can be observed that the proposed method outperforms all the comparison methods in terms of the average AUC across 16 projects, demonstrating its significant performance advantage. Compared to the homogeneous stacked ensemble models, the proposed method achieves an average AUC value higher by 22.79%, 5.35%, 16.8%, and 15.6% than the models with no feature selection, DT-based feature selection, SVM-based feature selection, and KNN-based feature selection, respectively. In the ablation study, the proposed method improves the AUC mean by 15.3%, 5.6%, and 1.9% compared to heterogeneous stacked ensemble models with DT+SVM feature selection, DT+KNN feature selection, and SVM+KNN feature selection. Furthermore, according to the Win/Draw/Lost statistics, our method achieved seven wins, one draw, and eight losses out of the 16 projects, with a significantly higher win rate than the other methods. Meanwhile, the Cliff’s Delta values indicate a substantial effect advantage of our method over the comparison models, especially when compared to the model without feature selection and the SVM-based feature selection model, with Cliff’s Delta values reaching 0.9531 and 0.8593, respectively. Cliff’s Delta is a non-parametric effect size measure used to quantify the amount of difference between two methods. The range of values is [−1, 1], with values closer to 1 indicating a large effect size. Typically, Cliff’s Delta is categorized into four effect levels, as shown in Table 13.

Table 13 Cliff’s delta and the effectiveness level.

Cliff’s delta	Effectiveness level	
0.000≤|δ| < 0.147	Negligible	
0.147≤|δ| < 0.330	Small	
0.330≤|δ| < 0.474	Medium	
0.474≤|δ| < 1.000	Large	

Finally, to validate the advancement of the proposed framework, we perform a comparative analysis with currently advanced machine learning models (Random Forest (Breiman, 2001), LightGBM (Ke et al., 2017), Natural Gradient Boosting (NGBoost) (Duan et al., 2020)) and two state-of-the-art defect prediction methods. The state-of-the-art methods include: Iterative Ensemble Genetic Classifier Algorithm (IEGCA), which combines genetic algorithms for feature selection and uses multiple binary classifiers along with an ensemble voting strategy for defect prediction (Ali et al., 2024); Nested-Stacking method, which improves the defect prediction model’s fitting ability and generalization ability by combining various boosting algorithms and custom deep learning algorithms based on the stacking criterion (Chen, Wang & Song, 2022). The experimental results are shown in Fig. 13 and Table 14.

Figure 13 Comparison of AUC among six methods.

Table 14 Homogeneous vs. heterogeneous stacked ensemble models for defect classification.

Data set	RF	LightGBM	NGBoost	IECGA	Nested-stacking	HEDSE	
Ant-1.7	0.7384	0.7134	0.7402	0.8418	0.7890	0.8389	
Camel-1.0	0.5103	0.4948	0.6836	0.9435	0.8072	0.9795	
Camel-1.2	0.6474	0.6118	0.6068	0.6648	0.6538	0.6899	
Jedit-4.0	0.6133	0.551	0.7034	0.7538	0.7238	0.7590	
Jedit-4.1	0.7587	0.7139	0.7399	0.8766	0.8393	0.8554	
Jedit-4.2	0.803	0.7828	0.7361	0.8444	0.8572	0.8926	
Jedit-4.3	0.5104	0.5079	0.7436	0.8609	0.8768	0.9827	
Log4j-1.0	0.6387	0.7354	0.737	0.9026	0.8245	0.9016	
Log4j-1.1	0.6547	0.6309	0.5476	0.9671	0.8128	0.9583	
Poi-2.0	0.6618	0.6758	0.6697	0.8874	0.8038	0.8724	
Poi-2.5	0.7381	0.7644	0.775	0.8954	0.8769	0.9151	
Poi-3.0	0.8309	0.8082	0.8366	0.8458	0.8259	0.8702	
Synapse-1.2	0.7477	0.7037	0.7127	0.8475	0.8121	0.8520	
Velocity-1.6	0.6663	0.6358	0.6358	0.8541	0.8116	0.8573	
Xalan-2.4	0.6960	0.7093	0.6920	0.7587	0.8414	0.7822	
Xerces-1.3	0.6286	0.6592	0.7359	0.8572	0.8274	0.8747	
Average	0.6777	0.6686	0.7059	0.8501	0.8114	0.8676	
Win/Draw/Lost	0/0/16	0/0/16	0/0/16	5/0/11	1/0/15	11/0/5	
Cliff’s Delta	0.9140	0.8984	0.8906	0.2314	0.5156	–	
Note:

Best values for each task are presented in bold.

From Fig. 13, it is evident that the proposed method achieves overall superior performance compared to RF, LightGBM, NGBoost, and two state-of-the-art algorithms, IECGA and Nested-Stacking. As shown in Table 14, the proposed method achieves an average AUC across 16 projects that is 18.98% higher than RF, 19.89% higher than LightGBM, 16.16% higher than NGBoost, 1.75% higher than IECGA, and 5.62% higher than Nested-Stacking. Furthermore, the Win/Draw/Loss statistics indicate that the proposed method achieved 11 wins, 0 draws, and five losses across the 16 projects, demonstrating a significantly higher win rate compared to other methods. Cliff’s Delta analysis further reveals that the proposed method provides significant improvements over RF, LightGBM, NGBoost, and Nested-Stacking, and a slight improvement over IECGA.

Conclusion and future works

To improve the accuracy of software defect prediction, this article proposes a software defect prediction classification framework. Firstly, a multi-strategy improved parrot optimization algorithm is proposed, and its effectiveness is demonstrated by applying the algorithm to typical test functions for optimization. Secondly, the multi-strategy improved parrot optimization algorithm is applied to the binary search space to form the BMEPO algorithm, which is then applied to the field of software defect prediction. Experimental results show that the BMEPO algorithm has good feature selection and generalization capabilities. Finally, to further improve the accuracy of software defect prediction classification, heterogeneous data stacked ensemble learning is proposed and compared with homogeneous data stacked ensemble learning models, demonstrating the effectiveness of our proposed method.

Although the BMEPO algorithm proposed in this study has achieved promising results in software defect prediction experiments, there are still some limitations. First, the BMEPO algorithm may suffer from performance issues when handling high-dimensional feature datasets. When the feature dimension is high or redundant features exist, the algorithm may require more time to converge and could potentially fall into a local optimum, which would affect its optimization efficiency and the stability of the results. Second, while datasets from the Promise open-source library are widely used in software defect prediction research, these datasets may not fully represent the feature distributions of data from other domains. Therefore, the adaptability and generalizability of the BMEPO algorithm on datasets from other fields still need further verification.

Future research could explore the application of the BMEPO algorithm in other domains, such as network intrusion detection and disease factor analysis. By optimizing feature selection, the detection capability of intrusion detection systems against various types of attacks could be enhanced, and key factors related to disease occurrence could be identified to improve the performance of prediction models for disease recognition. Additionally, although multiple baseline models have been used in the heterogeneous data stacking ensemble learning framework, the combinations of these models may not be optimal. Future work could investigate the exploration of all possible model combinations to find the best combination strategy, thereby constructing a more intelligent and automated prediction system to further improve prediction accuracy and stability. Finally, while this study primarily uses AUC as an evaluation metric to highlight the model’s performance under different classification thresholds during feature selection and model classification evaluation, future research could further explore the impact of other evaluation metrics (e.g., precision, recall, F1-score) on model performance to provide a more comprehensive assessment of the model’s adaptability and effectiveness across different datasets and tasks.

Supplemental Information

Supplemental Information 1 Defect analysis data.

Metric data of Java code and a label indicating whether it is defective.

Additional Information and Declarations

Competing Interests

The authors declare that they have no competing interests.

Author Contributions

Qi Fei conceived and designed the experiments, performed the experiments, analyzed the data, performed the computation work, prepared figures and/or tables, and approved the final draft.

Guisheng Yin conceived and designed the experiments, authored or reviewed drafts of the article, and approved the final draft.

Zhian Sun conceived and designed the experiments, authored or reviewed drafts of the article, and approved the final draft.

Data Availability

The following information was supplied regarding data availability:

The raw measurements are available in the Supplemental File.

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
