# Peer review of "Feature selection using a multi-strategy improved parrot optimization algorithm in software defect prediction"

_PeerJ Computer Science, doi:10.7717/peerj-cs.2815_

## Round 0.1 · original submission · Major Revisions

Please carefully improve the work according to the comments. Then is will be sent to the reviewers to be evaluated again.

·

Basic reporting

The paper presents a novel, well-structured, and practically significant methodology with clear experimental results. It is suitable for publication with minor changes to enhance clarity and comprehensiveness.

Experimental design

No Comment

Validity of the findings

No Comment

·

Basic reporting

1. A few grammar corrections, spelling error, and acronyms must be looked at carefully throughout the manuscript
2. Equations citations should be done properly

Experimental design

Original primary research within the Aims and Scope of the journal.
The research question is well-defined, relevant, and meaningful. It states how research fills an identified knowledge gap and is well-demonstrated.
In the Results and discussion, details of the implementation environment and overview of the dataset are required

Validity of the findings

All underlying data have been provided; they are robust, statistically sound, & controlled.
Explanations for the dataset should be given
Conclusions are well stated, linked to original research question & limited to supporting results

Additional comments

First and foremost, I would like to appreciate the authors for the good and extensive work in evaluating the performance of their proposed model. The paper was organized well. The following suggestions would help the readers to enhance the readability:

1. The abstract is not clear. For instance, lines 17 to 19 could be rewritten. And what does the phrase” most datasets convey” mean? Furthermore, the abstract should answer the question: What is the significance of your work for researchers in the same field?”
2. In line 29, give valid reference for the Pareto principle
3. In line 38, “defectsSong et al.” leave a space between these words, and check for the same while citing references throughout the manuscript
4. All the acronyms should be abbreviated on their first usage, and the acronym should be referred to on its subsequent usage. Some of the acronyms have been abbreviated multiple times and few weren’t. Eg: line no. 234,238, 240, 241, 304,311 etc..
5. In line 48, change “The paper” to “This work” or “This paper”
6. In the significant contributions presented in the introduction section, there are some repeated statements from the previous paragraph:” Our proposed model has proven to outperform homogeneous data stacking ensemble learning models.” Also, in some places, you have used this model, and in other places, you have used our proposed model, so maintain uniformity in this usage.
7. What is this “BMCPO”?? If this refers to your proposed model, abbreviate it and use it throughout the paper. This would improve the readability
8. In line no.91 “Arar et al.Arar and Ayan (2015)” check this, and same kind of typo errors with reference management software occur at different places (For instance, in line no: 98, 102,104)
9. In line 120, what does “ each individual” refer to? And PAO can be explained simply to enhance readability and understandability
10. While using comma, leave a space after that: line no 124 – 127
11. Some special characters were inserted in line no.129
12. In stacked ensemble learning: line nos: 135 to 137, predictions + input data is fed as input to metamodel (as per explanation), but it is not reflected in Figure 1. Will the meta model in a stacked ensemble take only predictions as features or the input data? Clarify
13. In 3.2, 3.2.2, 3.2.3 capitalise the first letter of the section title
14. Why is the chaotic state parameter set to 0.5??
15. Usually equation is denoted as Eqn. check this in line no. 193
16. Change the 4.1 section title to a Result Analysis of MEPO and Experimental Comparison with other Models. Similarly, change 4.2, 4.3
17. In line no. 249, it is Table 5 I think, change it
18. In table 10. Check the column names, and NGBoost was not abbreviated
19. Why were specific models chosen for stacking, the justification for this is missing
20. In the Results and discussion, details of the implementation environment and overview of the dataset are required

Reviewer 3 ·

Basic reporting

• The manuscript is well-structured, with a logical flow from introduction to conclusion. The sections are clearly and appropriately titled.
• The language used is clear and precise, despite minor grammatical errors.
• The literature review provides sufficient context for the problem and appropriately references relevant studies, such as existing feature selection methods and ensemble learning techniques.
• Figures and tables are relevant and well-labeled. The inclusion of diagrams for algorithms and frameworks aids comprehension. However, Figure resolutions should be improved for clarity.

Experimental design

• The objectives of the study are well-defined, with a focus on improving software defect prediction using feature selection via a novel multi-strategy enhanced parrot optimization algorithm (MEPO).
• The methodology is robust, employing standard metrics (AUC) for performance evaluation and multiple comparative baselines (e.g., BPO, BWO, BCPO).
• The dataset preprocessing steps, including SMOTE for addressing class imbalance, are adequately detailed. However, additional explanation on why certain datasets were chosen (e.g., PROMISE repository datasets) would strengthen the rationale.
• The parameters of the algorithm are clearly described, but sensitivity analyses for key parameters like mutation rate or population size are missing.

Validity of the findings

• The findings are well-supported by experimental results, demonstrating the superiority of MEPO in optimization tasks and feature selection for defect prediction.
• Statistical analysis is sufficient; standard deviation and multiple independent runs ensure reliability. However, the manuscript lacks confidence intervals or statistical tests to compare algorithm performance rigorously.
• The heterogeneous data stacking ensemble (HEDSE) framework’s results outperform homogeneous models, providing a strong argument for its use. Yet, an ablation study isolating the effect of HEDSE would strengthen claims.

Additional comments

• The manuscript makes a valuable contribution by integrating a novel optimization algorithm with ensemble learning for a critical problem in software engineering.
• The writing can benefit from minor revisions for fluency and consistency. For example, terms like "heterogeneous data stacking ensemble" should be abbreviated consistently throughout.
• The introduction sets the stage well but could more directly highlight the practical implications of defect prediction improvements.
• Future work directions are insightful but would benefit from more specific suggestions, such as real-world deployment challenges or computational cost comparisons.

Specific Suggestions for Improvement:

1. Include a sensitivity analysis for MEPO parameters.
2. Provide statistical tests for performance comparisons.
3. Improve the quality of figures and ensure consistent formatting throughout.
4. Revise certain sentences for grammatical clarity and conciseness.
5. Expand on practical implications and limitations in the discussion.

Reviewer 4 ·

Basic reporting

The paper is well written and relatively easy to follow. The quality of English is satisfactory. Figures are of adequate quality and the data set is supplied.

Experimental design

The research question is formulated well. The authors propose a novel model for defect prediction and compare it to some of the existing ones.

Validity of the findings

There are several issues with this paper as it appears that the authors are trying to add a lot of content while not clearly reporting all the necessary details. Firstly, it is not clear how the MEPO and PO differ when it comes to computational efficiency. Some results regarding this should be given. Secondly, BMEPO, the purposed algorithm at the center of the stated contributions, is never clearly defined in the paper. The authors need to clarify all the metrics used during evaluation.

---

## Round 0.2 · accepted · Accept

I‘m happy to inform you that this version satisfied the reviewer successfully. I believe it can be accepted currently. Congrats!

·

Basic reporting

No comment

Experimental design

No comments

Validity of the findings

Validated well as suggested by the reviewers

Additional comments

The authors have incorporated all the suggestions specified and can be accepted

Reviewer 4 ·

Basic reporting

The paper is well written and easy to follow. It conforms to the journal's standards.

Experimental design

The research is within the aims of the journal.

Validity of the findings

The research findings are valid.

Additional comments

The authors have addressed my concerns and the paper can be accepted.